# The assessment of physical risk taking: Preliminary construct validation of a new behavioral measure

**Edward A. Smith**, **Stephen D. Benning***

Department of Psychology, University of Nevada, Las Vegas, Las Vegas, Nevada, United States of America

* stephen.benning@unlv.edu

**Data Availability Statement:** All data and analytic scripts can be found in the Open Science Framework at https://osf.io/a85c2/. We also preregistered our analytic plan at https://osf.io/

## Abstract

Risk taking is a complex heterogeneous construct that has proven difficult to assess, especially when using behavioral tasks. We present an exploratory investigation of new measure–the Assessment of Physical Risk Taking (APRT). APRT produces a variety of different outcome scores and is designed as a comprehensive assessment of the probability of success and failure, and magnitude of reward and punishment of different types of simulated physically risky behaviors. Effects observed on the simulated behaviors are hypothesized to reflect similar effects on real world physical risks. Participants ($N = 224$) completed APRT in a laboratory setting, half of whom had a 1.5 s delay interposed between button presses. Exploratory analyses utilizing generalized estimating equations examined the main effects and two-way interactions among five within-subject factors, as well as two-way interactions between the within-subject factors and Delay across four APRT outcome scores. Results indicated that Injury Magnitude and Injury Probability exerted stronger effects than any of the other independent variables. Participants also completed several self-report measures of risk taking and associated constructs (e.g., sensation seeking), which were correlated with APRT scores to assess the preliminary convergent and divergent validity of the new measure. After correcting for multiple comparisons, APRT scores correlated with self-reported risk taking in thrilling, physically dangerous activities specifically, but only for those who did not have a delay between APRT responses. This promising exploratory investigation highlights the need for future studies comparing APRT to other behavioral risk taking tasks, examining the robustness of the observed APRT effects, and investigating how APRT may predict real-world physical risk taking.

## Introduction

Risk taking is a heterogeneous construct used in many areas of research. A common theme in the risk taking literature revolves around engaging in behaviors or decisions that involve multiple possible outcomes, some of which are negative [1]. Some theorists measure risk only as a function of the magnitude or probability of these negative outcomes, with the type of risk sometimes included as a moderating factor [2]. Indeed, a number of definitions of risk focus

**Funding:** The authors received no specific funding for this work.

**Competing interests:** The authors have declared that no competing interests exist.

exclusively on potential negative consequences [3]. Both playing poker and driving drunk involve making decisions that have potentially negative outcomes (e.g., losing money, crashing, being arrested). However, some fields define risk as general outcome variance, which encompasses situations in which all outcomes are neutral or positive, or even always positive [4]. An example of this would be studies on expected utility theory, which often use monetary gambles or similar paradigms that focus solely on gains [5]. In these cases, the magnitude or probability of positive outcomes is central to decision making. Furthermore, the magnitude of both negative and positive outcomes influence risky decision making in many situations. For example, experienced gamblers understand when they have a good chance of winning a round of their game of choice and can alter their betting based on their perceived chance of success [6]. Similarly, people who have had a large number of alcoholic drinks are more aware of their impairment and thus choose to drive less often than individuals who have had a medium number of drinks, reducing their risk of crashing their vehicle [7].

In behavioral economics, the widely influential prospect theory is rooted in observed patterns of differential responding to changes in both loss and gain probabilities and magnitudes in an individual's specific situation [8]. Furthermore, those who are high in sensation seeking, impulsivity, and other risk-related personality constructs tend to overestimate the extent of potential rewards of risky behaviors and thus exhibit below average loss aversion [9, 10]. Recent research has also suggested the existence of a generalized risk preference personality trait (*R*) that is analogous to the *g* factor in the intelligence testing literature, predisposing people to engage in a variety of risky behaviors [11]. Thus, the likelihood of *both* success and failure, as well as the magnitude of each, factor into the decision to engage in a variety of risky behaviors, with the strength of their influence varying according to each individual's unique risk preference and to a lesser extent the specific domain of risk taking (e.g., physical, financial, social).

These four factors (the separate probabilities of loss and gain, and the separate magnitude of each) are key components that affect risky decision making, and each are associated with different, albeit related, neurobiological and cognitive processes [12, 13]. These four factors may also interact with different risky behaviors, as the strength and extent of the effects of these factors on risk taking may vary between specific risky behaviors. For example, the magnitude of reward is likely to play a larger role in behaviors such as gambling, where individuals are tempted to risk losing more money for the chance to win a large sum. In contrast, an intoxicated individual may be more concerned with the likelihood of being caught or crashing their vehicle (i.e., injury probability) if they were to attempt to drive. Thus, a comprehensive behavioral task should examine all five risk taking factors: the effects of the four underlying influences on decision making and the specific risky behavior being examined. Nevertheless, few tasks operationalize these factors simultaneously within behavioral assessments of risk taking in psychological studies. Beyond that, increasing evidence suggests that these tasks do not correlate well with real-world risk taking measures [4, 14], indicating a need for more complex behavioral tasks that parse these underlying factors of risky decision making and correlate with self-report measures of specific forms of risk taking.

## Common behavioral measures of risk taking

The most commonly used behavioral measures of risk vary in their coverage of these aspects of risk taking. One of the most popular behavioral measures is the Balloon Analogue Risk Task (BART), a computerized task in which participants earn money (often imaginary) by continually inflating a virtual balloon [15]. Participants may stop inflating a balloon at any time to "bank" their current earnings and start a new balloon trial. Successive pumps increase the risk

of the balloon exploding, which causes the participant to lose any earnings from that trial. Though the task manipulates the probability of loss as part of the task (as the likelihood of the balloon popping increases with each pump, and the starting probability varies between trials in most versions of the test), its ability to manipulate other factors of risky decision making is more limited. The amount of money earned with each pump does not vary within or across trials, and the probability of earning money on a pump is simply the inverse of the probability of the balloon popping (i.e., 1 minus the loss probability). There is no neutral outcome in which money is neither earned nor lost, thus making the probabilities of gain and loss two ways of examining the same variable. Participants augment a trial's reward magnitude with each successful pump in a trial, but the amount each pump increases that reward remains constant throughout the task.

The relative importance of gains and losses has been modeled as a function of participant behavior on the BART, accounting for subjective perception of reward sensitivity and loss aversion [16]. Objective manipulations of loss (explosion) probability also affect subjective perceptions loss probabilities in the BART, finding that participant perceptions of loss probability deviate greatly from the actual probabilities, but that this perceived probability was more closely associated with their actual decisions within the task and external risk taking [17]. Unfortunately, manipulating factors beyond loss probability and reward amount in the BART itself is challenging (e.g., loss magnitude cannot be altered much as the balloon either explodes, which results in loss of money, or it does not and more money is earned). Moreover, recent research has also criticized the representative design of the BART, suggesting that "fixing" or altering the task in some way to improve it may not be the best path forward to study risky decision making and the factors influencing it [18]. Instead, new tasks that inherently possess high representative design should be created and implemented.

Another popular behavioral risk taking task is the Iowa Gambling Task [19]. In the IGT, participants select a total of 100 cards by choosing from 4 decks, with no more than 60 cards being allowed to come from the same deck. Each card gives or takes away a certain amount of "money" to the participant. The decks are arranged such that two give an overall net positive gain (advantageous) but lower individual "win" values, and the other two give an overall net loss (disadvantageous) but with larger individual "win" values [20]. Scores for decision making are calculated by subtracting the number of disadvantageous deck choices from the number of advantageous deck choices. Most individuals typically show a reversal learning effect midway through the task, at which point they recognize that the disadvantageous decks are now overall negatively impacting their overall performance (as their larger "win" amounts become less frequent) and consciously switch to the net positive, but smaller rewarding, advantageous decks [21].

This learning effect (or lack thereof, in the case of neurologically impaired individuals) has been hypothesized to be the cause of group differences in IGT performance, as opposed to differences in risk taking propensity [22], calling into question the construct validity of the task. Still, with respect to its manipulation of the various elements of risk, the four decks in the task do have varying magnitudes of both rewards and penalties, as well as differing reward and penalty probabilities [20]. However, examining each of these effects individually would be difficult, as the task is essentially one very long trial, with the variations in each of these factors being controlled by the participant (via the deck they choose for each of their 100 selections) and not the experimenter. Still, efforts have been made to decompose the task into other factors, such as the impact of losses and gains on option evaluations, the rate the learning effect in the task is acquired, and the consistency of the learning effect [23]. As with BART, these models focus more heavily on participant perceptions of the task, rather than intentional objective

manipulations of probability by the task itself. Thus, tasks that allow for manipulations of these factors that are not influenced directly by the participant are necessary.

The Columbia Card Task (CCT) explicitly manipulates reward magnitude, loss magnitude, and loss probability within a task in which the participant turns over virtual cards [24]. It also assesses "hot" and "cold" risk situation processing by revealing feedback after each card flip or waiting until all cards are flipped. However, to the extent CCT performance correlates with any personality constructs [25], it appears to be more related to reward sensitivity than impulsivity or physical risk taking [26, 27]. Furthermore, CCT performance's relationships to BART and IGT performance have been non-discernible [24, 28]. There is consequently a need for specialized behavioral tasks that not only assess the different cognitive constructs and situations that comprise risky decision making, but that also still show appropriate correlations with self-report measures of risk-related constructs. For instance, the Sensation Seeking Scale's Thrill and Adventure scale, the UPPS-P impulsivity measure's Sensation Seeking scale, and the Domain Specific Risk Taking measure's Recreational scale assess physical risk taking specifically. These measures are distinct from the externalizing antisocial behavior and substance misuse that are associated more with boredom susceptibility and other impulsive stimulus seeking patterns [29].

Our study bridged this gap in the literature by introducing a specialized type of behavioral task that incorporates imagery of actual physically risky activities, as opposed to a novel game that does not have a real-world equivalent (such as selecting cards or pumping up balloons for money). Though no behavioral measure of risk taking can ethically simulate most risk taking behaviors that entail physical injury, we depicted multiple real-world behaviors in a task to predict more accurately patterns of physical risk taking and risk-related constructs.

## The Assessment of Physical Risk Taking (APRT)

APRT is a computerized task comprising 64 unique trials that are representative of physically risky activities. During each trial, participants choose whether to proceed with a risky activity displayed by a picture (with accompanying text description) to earn points and avoid losing health. Choosing to proceed results in one of three outcomes: gaining points, gaining no points but also not losing health, or losing points and health. This tri-outcome format is distinct from other popular behavioral tasks, which typically have only two outcomes (loss or gain). Previous research on three-outcome gambling tasks has suggested that participants evaluate trial outcomes primarily as function of a reward magnitude x reward probability interaction, and that salience plays a role in prioritizing these factors over others, such as loss probability [30].

Additionally, many of the typical assumptions within ordinary prospect theory, which applies best to two-outcome gambles, do not extend in more complex gambling paradigms; rather, alternative models (e,g., cumulative prospect theory) using a "rank-dependent" strategy explain behavior better [31, 32]. Under these models, different elements of a risky decision (e.g., reward/loss probability/magnitude) serve to rank each possible outcome, with these ranks determining the overall value of each risky decision. However, the apparent importance of reward-related variables may be due to the monetary gambling nature of the tasks examined in multi-outcome risk taking research, in which the chance to win money is more salient and weighed as more significant than the risk of losing money. In situations where there is a more severe risk in cases of loss to the subject (i.e., physically dangerous activity), loss probability and magnitude may instead have a stronger influence on perceptions of risk. Thus, APRT provides a non-monetary gambling risk taking task that allow for examination of the influence of these elements of risky decision making.

APRT manipulates five elements of risk: Reward Probability (low/high), Reward Magnitude (low/high), Injury Probability (low/high), Injury Magnitude (low/high), and Picture Type

(encountering a dangerous animal, standing/walking along a steep cliff face, attempting to photograph a disaster, or attempting to rescue someone in physical peril). Four outcome scores derive from APRT performance: number of Points, Go Presses, Injuries, and Remaining Health. The convergent and divergent validity of these manipulations and outcome scores can be tested with self-reports of physical risk taking and other constructs related to externalizing behavior. If the APRT is a representative measure of physical risk taking, it should relate more to measures of thrill and adventure seeking and recreational risk taking than of risk perception. If the APRT measures physical risk taking specifically rather than a broader behavioral disinhibition, it should not be related to other forms of sensation seeking, antisocial behavior, or substance use.

However, interposing delays between each APRT response should reduce its representation of the ever-heightening thrill of physical risk taking across rapid key presses [18]. To test this notion, we developed "hot" and "cool" versions of APRT. The "hot" version updates feedback immediately after each button press and allows participants to proceed as quickly as they wish. The "cool" version enforces a 1500 ms delay between key presses in which the points and health are updated. During the delay, they are unable to continue to their next key press decision. The introduction of delays during risk taking tasks, such as card games, has been shown to reduce maladaptive preservations in task behavior in participants with personality traits related to high levels of risk taking [33]. The "cool" condition conceptually resembles the "cold" version of the CCT, as it withholds performance feedback for a certain period, albeit only for a brief delay as opposed to an entire trial. The "hot" and "cool" APRT conditions may show different patterns of responding and relationships to personality traits, as is seen in the CCT [26]. We therefore examined whether a delay impacted APRT performance and reduced APRT scores' relationships with personality traits related to risk taking.

## Aims and hypotheses for the current study

The first aim of this study explored how the APRT functioned as a measure of physical risk taking. We examined the degree to which the proposed factors underlying risky decision making (e.g., reward magnitude, injury probability) affected the various APRT scores. We hypothesized that factors relating to injury would affect participants' scores more strongly than factors associated with reward or picture category. Though reward plays a more important role in three-outcome monetary gambles, physical risks entail more severe "losses" (e.g., bodily injury, illness), so we suspected that the role of the injury variables would be stronger than reward due to increased salience of the consequences of loss [30]. We believed these to be the most important effects that would emerge, but we explored all main effects and two-way interactions to characterize thoroughly the easily interpretable combinations of the APRT's manipulations.

The second aim of the study examined the convergent and divergent validity of APRT scores by comparing them to self-report measures of risk taking and related variables. In particular, we hypothesized that APRT scores would correlate with self-report measures of risk taking but not risk perception or other risk taking related constructs (e.g., externalizing symptoms, religiosity), due to APRT's specific focus on physically dangerous activities. Both of these aims used stringent controls for false positive test results to maintain rigor in our exploratory analyses.

Finally, the third aim of the study determined whether an enforced delay during APRT (the "cold" version) discernibly altered the relationships of the APRT scores to the self-report measures. We hypothesized that the "cold" version of APRT would attenuate the relationships between participants' APRT performance and their self-reported levels of risk taking compared to the "hot" version, as is the case in the CCT [26, 33].

## Method

### Participants

Participants were 224 students recruited through the psychology subject pool at the University of Nevada, Las Vegas. The UNLV Institutional Review Board approved the methods and consent procedures of this study (protocol #711255–5). Participants received written consent and debriefing, and they were given the opportunity to ask clarifying questions before and after the study. They received class credit as compensation. The sample was 71% female and had a mean age of 19.71 ($SD$ = 3.03, range = 18–42). Of the 90.1% of participants who reported their race, 40.1% were White, 38.5% Asian, 13.9% African American, 6.4% Pacific Islander, 1.2% Native American; across all these participants, 35.7% endorsed Hispanic ethnicity. A power analysis revealed that a sample size of 281 would have been required to observe the smallest discernible main effect size found in an unpublished pilot study ($d$ = 0.167) at the recommended power level of .80. However, the next smallest main effect size ($d$ = 0.232) only required a sample of 202. Thus, we had a sufficiently powered sample to detect effects of similar size to the pilot study.

### Measures

**Assessment of Physical Risk Taking (APRT).**   As described above, APRT is a computerized task in which participants complete 64 unique trials. The APRT is available at https://osf.io/4vm7k/, and this study used version 4.2 of the program. Each trial involves the participant viewing a picture that depicts a first-person view of a physically dangerous activity from one of four categories (encountering a dangerous animal, attempting to photograph natural disasters, attempting to help someone in physical danger, and standing on the edge of a high ledge or cliff). These pictures represent real-world activities in which an individual high in physical risk taking propensity might engage.

At the start of the trial, the picture is accompanied by text along the top of the screen describing what the participant is doing (e.g., "You are attempting to photograph natural disasters" would accompany a picture of a tornado). The screen also shows the participant's current health and points scores for that trial. Each trial begins at 100 health and 0 points.

During each trial, participants choose whether to proceed with the displayed activity to try and earn points. Choosing to proceed (by pressing the "p" key on a standard keyboard) results in either gaining points and a reward message that replaces the initial activity description ("Lucky you–points awarded!"), gaining no points but also not losing health ("Sorry, no points awarded"), or losing both points and health ("Oh no! You have been injured–choose again"). The condition with no change in points was implemented to separate the probabilities of rewards and injuries, as opposed to tasks like BART where one is simply the inverse of the other [15].

Points and health are updated after each button press. If the participant's health drops to zero, a "death" screen is displayed, in which the entire screen changes to only display the following message: "You have died. You have lost all of your points for this trial. Please wait for a few seconds while the computer resets for the next trial." Choosing to quit the current trial (by pressing the "q" key) allows the participants to save all their current points and safely end the current "activity", automatically proceeding to the next trial (with a new picture, text description, and reset health and points). Fig 1 shows an example of how a trial may unfold for the participant.

The probabilities of reward or injury, as well as the relative ranges of possible rewards and injuries, vary across trials but are consistent throughout a trial. Table 1 summarizes the exact

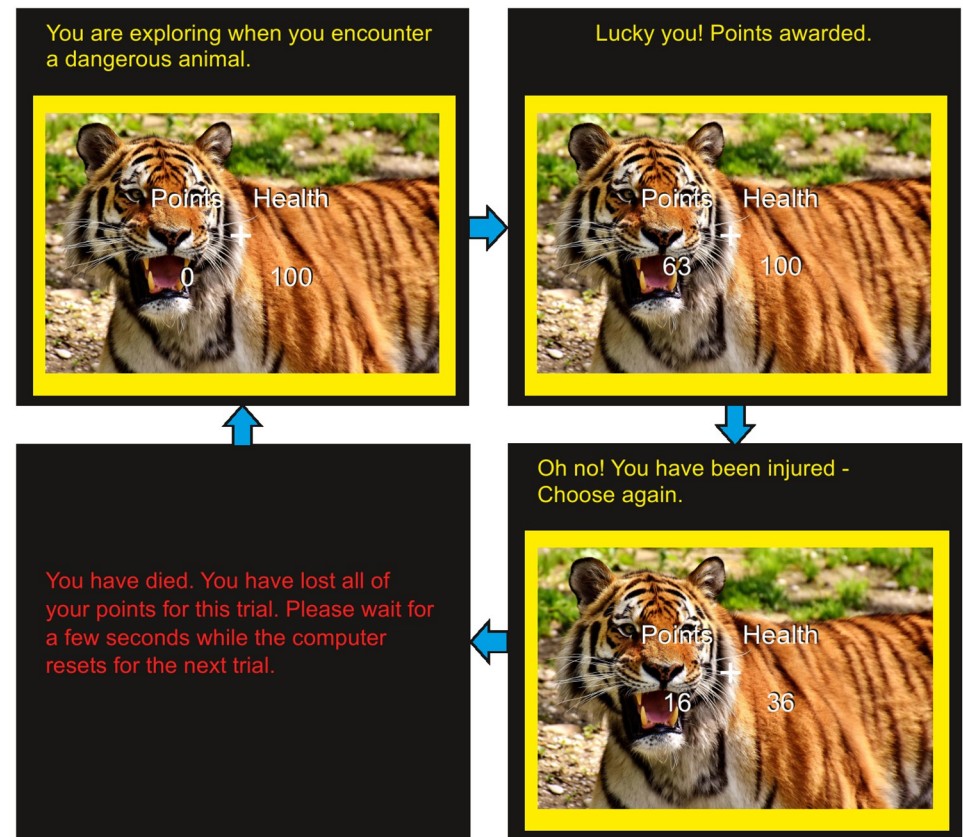

**Fig 1. APRT trial example.** Top left: the participant begins a new trial with 0 points, 100 health, and an image (with description) of a physical risk-taking activity. The yellow border color indicates a high probability of injury (versus a blue color indicating a low injury probability); the solid border color indicates a high probability of being rewarded (versus a checked border indicating a low reward probability). Border color and solidity only changes at the start of new trials, and the magnitude of reward and injury is revealed only through proceeding through the trial and experiencing each outcome. Top right: the participant has pressed the proceed ("p") button and won 63 points as a result. Bottom right: another "p" press has caused an injury, and thus the participant lost 64 health and points. Bottom right: another "p" press has caused an injury severe enough for health to drop below 0, so the participant is shown a "death" screen and loses any points gained during this trial. If the participant does *not* "die" after a "p" press, they may press the quit ("q") button at any time to save all points earned on the trial to their grand total (not displayed) and proceed to the next trial. The trial's points are reset to 0; the health to 100; and a new picture, border, and description are displayed.

probabilities for the different reward and injury conditions, as well as the ranges of health and points lost per injury and points gained per reward. Participants are informed that the border pattern surrounding each trial picture indicates the reward probability of that trial (solid = high reward probability, striped = low reward probability), whereas the color of the border indicates the injury probability (blue = low risk of injury, yellow = high risk of injury).

**Table 1. Summary of reward/injury probabilities and severities.**

| Variable | Low | High |
|---|---|---|
| Injury Magnitude (per injury) | Randomized range: 5–35 | Randomized range: 60–90 |
| Reward Magnitude (per reward) | Randomized range: 5–25 | Randomized range: 50–250 |
| Injury Probability | 1/15 | 1/5 |
| Reward Probability | 2/3 | 9/10 |

This procedure protects the task from learning effects present in other behavioral measures of risk taking, such as the IGT [22]. By informing the participants of the trial-level probability manipulations, task performance is not confounded by differences in probabilistic learning. Participants' behavior during the task is therefore more indicative of their actual risk taking propensity rather than their ability to learn or adapt to the task. Furthermore, neurobiological research has shown that there is stronger activation of brain areas associated with risk perception when outcome probabilities are indicated compared to when the probabilities are left ambiguous [34]. Thus, participants may use these probabilities when making decisions in the task more readily than if they had to infer them from their own performance. Participants are not given cues about the magnitude of rewards or injuries in each trial (i.e., how many points they could win or health they could lose from a single press). Instead, this information is displayed through the outcomes of each button press in a trial.

Participants are shown previous high scores of other individuals who have completed APRT (which are real scores taken from pilot studies using a previous APRT version). They are told that they will see their total score after the final trial and that they should try to earn as many points as possible. There are not given any instructions regarding health, save for a warning about the loss of all points earned that trial should their health drop to zero. Participants complete four practice trials (one from each picture category) to become accustomed to the task before the actual 64 trials begin. During each trial, participants only see the number of points they could win during that trial itself (as shown in Fig 1). The task terminates after a display of the total number of points earned across all 64 trials.

During each trial, participants choose whether to proceed with a risky activity displayed by a picture (with text description) to try to earn points and avoid losing health. Choosing to proceed results in one of three outcomes: gaining points, gaining no points but also not losing health, or losing both points and health. Five variables are manipulated to make each trial unique: Reward Probability (low/high), Reward Magnitude (low/high), Injury Probability (low/high), Injury Magnitude (low/high), and Picture Type (encountering a dangerous animal, standing/walking along a steep cliff face, attempting to photograph a disaster, or attempting to rescue someone in physical peril). S1 Table illustrates the number of trials administered for each level of each of these factors, as well as every 2-way interaction combination among them.

As mentioned previously, other risk taking tasks lack a representative design that poorly reflects the environment in which risk taking occurs, such as the BART providing a poor simulation of a risky behavior in its balloon-pumping methodology [18]. We thus designed the APRT stimuli to reflect photorealistic instances of risk taking in a dangerous environment (e.g., helping someone in a manner likely to harm oneself, proximity to dangerous animals). Though these categories may not reflect everyday forms of physical risk taking the average person encounters, the severity of the circumstances serves to maximize individual differences (and therefore effect sizes) in task performance while still providing improved ecological validity compared to current risk taking measures. For instance, encountering a bear may be a rare and unlikely event, but it is still a realistic form of physically dangerous risk compared to the complete lack of perceived danger in popping a pumped balloon. Thus, though the APRT stimuli may represent uncommon forms of physical risk taking, their design is more representative than extant measures such as the BART.

Four outcomes scores derive from APRT performance, which are calculated both as totals and as the average difference in the score between each of the low/high variable conditions (with Picture Type being split into Animal-Cliff and Hero-Disaster contrasts): number of Points, Go Presses, Injuries, and Remaining Health. The latter two outcomes are secondary to the first two, as is evident by how the current results are presented below. Points and Go Presses are direct measures of participant engagement in the task (i.e., participants are

explicitly instructed to maximize points, which must be accomplished through higher Go Presses). In contrast, Injuries and Remaining Health indirectly assess how often participants tended to persevere as risk increases within a trial (i.e., health becomes closer to zero in the trial, each injury increases the chance the next will result in loss of all points in that trial). As participants are not instructed to maximize health or minimize injuries, these outcomes (especially health, as it is displayed to them) serve more as a way of informing participants of the rising risk of continuing with a trial if they persevere after "losing" (i.e., receiving an injury and losing health) at least once. The displayed health score also helps participants understand that a single loss does not end a trial, allowing for a better representation of the continuous dimension of risk taking propensity.

Descriptive statistics and Cronbach's α internal consistencies for these outcome scores are in S2 Table. Descriptive statistics for all self-report measures are in S3 Table.

**UPPS-P impulsivity measure.** The UPPS-P is a 59-item inventory that assesses separate personality facets that are meant to represent the construct of impulsivity [35]. The UPPS-P consists of five subscales: negative urgency, positive urgency, lack of premeditation, lack of perseverance, and sensation seeking. Responses are indicated using a 4-point Likert scale ranging from 1 (*disagree strongly*) to 4 (*agree strongly*).

**Short Drug Abuse Screening Test (SDAST).** The SDAST is a 20-item, true/false measure that assesses the severity of drug use and the degree of any problems related to the drug use occurring in the past year [36].

**Behavior Report on Rule Breaking (BHRQ).** The BHRQ assesses adolescent and adult antisocial behavior. It consists of 33 items adapted from other measures and structured interviews [37–39], to which participants indicate how often or how many times they have engaged in a given behavior (e.g. "How often have you seriously hurt someone other than your brother or sister?"; during both in adulthood and childhood. This measure's subscales include Child Aggressive behavior, Child Non-aggressive behavior, Adult Aggressive behavior, and Adult Nonaggressive behavior.

**Alcohol Dependency Scale (ADS).** The ADS is a 29-item measure that assesses the severity of an individual's reliance on alcohol (e.g. urge to drink, tolerance, withdrawal symptoms; [26]). Response options change from item to item, with total scores ranging from 0–47. Higher scores indicate heavier dependence.

**Domain Specific Risk Taking (DOSPERT).** The DOSPERT is a 40-item measure that assesses risk behaviors, as well as perceived risk, across a variety of domains (social, ethical, health/safety, recreational, and financial; [40]). Participants responded to each item using two 5-point Likert scales: how likely they are to engage in the risky behavior (1 "*very unlikely*" to 5 "*very likely*") and their own perceived risk level of the behavior (1 "*not at all risky*" to 5 "*very risky*"). Subscale internal consistency scores for each of these ranged from .52 to .84 (Behavior) and .56 to .74 (Perception).

**Sensation Seeking Scale (SSS).** The SSS consists of 40 items that assess an individual's preference for high-arousal, novel activities over low-arousal, conventional ones [41]. The SSS consists of four subscales: thrill seeking, experience seeking, disinhibition, and boredom susceptibility. Items consist of forced-choice responses between two options, with higher scores being indicative of greater sensation-seeking tendencies.

**Sex, Health, and Automotive Risk Taking (SHART).** The SHART is a new measure of risk taking behavior and risk perception that attempts to assess more real-world domains of risk: sex, health, and automotive use. It consists of 36 questions, which are aggregated into three subscales, one for each domain. Participants respond to each question with how often they have engaged in a risk taking behavior (e.g., had sex, broken the speed limit while driving, smoked cigarettes) in the past 6 months using a 5-point scale ("Never," "Once a month or

less," "About once a week or two," "A few times per week," "Nearly every day") with higher scores indicating higher levels of risk taking behavior. Previous research in our lab established that the health subscale did not represent a unitary construct with reasonable internal consistency. Somewhat similar findings occurred in the current sample as well ($\alpha$ = .63 and .78 for risk taking and perception, respectively), so the health scale was dropped from subsequent analysis here as well.

**Centrality of Religion Scale (CRS).**   The CRS is a 15-item scale that measures several dimensions of religiosity and religious behavior [42]. Respondents use 5-point Likert scales to indicate how often they engage in religious behaviors (e.g., attending services, praying), the extent to which they hold certain religious beliefs (e.g. "To what extent do you believe in an afterlife?"), and how interested they are in religious issues/questions, both within and outside their faith.

## Procedure

Participants completed all the self-report measures except the CRS in the order listed above, then the APRT, then the CRS on computers located in a university laboratory setting as part of a larger study on personality and behavior. The entire study was programmed in PsychoPy, and its 64 trials were counterbalanced across all 5 independent variables. Participants with an even number in this study were placed into the 1500 ms Delay condition, whereas those with an odd study number had no delay interposed between responses. A trained research assistant sat participants at one of five lab computers, gave each participant the opportunity to ask questions during the consent process, and monitored participants during the study. All participants received a verbal debriefing once the study procedure was complete. Most participants completed the study within two hours.

## Data analysis

**APRT main effects and interactions.**   To model the effects of the independent variables on the four different APRT dependent variables, we used a series of generalized estimating equations (GEEs). We created a separate GEE for each dependent variable (Winsorized at +3 $SD$) to model the main effects and interactions of the 6 independent variables. We estimated all main effects, two-way interactions among the 5 within-subject factors, and interactions between within-subject factors and Delay (the sole between-subject factor) assuming an unstructured estimated covariance matrix. Due to the skew in Go Presses and Injuries scores, we performed a log transformation on these scores prior to running each GEE to retain statistical power that would have been otherwise lost from assuming the unstructured covariance matrix.

We decomposed statistically discernible main effects for Picture Type and interactions using *post hoc* pairwise comparisons utilizing a sequential Sidak (also known as Holm-Sidak) correction to control for multiple comparisons. This method resembles a sequential Bonferroni correction in method but employs a more complex formula to determine the critical $\alpha$ for each comparison to maintain the false positive rate precisely at $\alpha$ [43]. A nominal critical $\alpha$ of .05 was used for all analyses, and we conducted all analyses in SPSS version 25.

**Correlations with risk taking and other self-reports.**   We calculated Pearson correlation coefficients to examine the relationships between APRT difference and total scores and the self-report measures. Though difference scores have been criticized as quantifications of repeated measures effects, psychometric research has elucidated two characteristics that enhance the utility of difference scores [44]. High reliabilities of each component of a difference score both creates more reliable difference scores and controls false positive and false

negative error rates when using difference scores [45]. The reliability coefficients for the individual high and low conditions within each factor across all scores were high to nearly perfect (α range: .72-.97, *M*: .93, median: .95), bolstering the utility of APRT difference scores. Additionally, difference score components whose variances differ substantially across the two levels of a factor are particularly reliable [46], which was the case for most factors across APRT outcomes (especially for Points and Go Presses). Thus, difference scores were well-powered and error-controlled measures in this experimental context [47, 48].

We applied separate sequential Sidak corrections to each family of correlations to correct for multiple comparisons within that family. At https://osf.io/bd5p6/, we postregistered hypotheses about which APRT difference scores would be convergently and divergently related to self-report measures in our dataset after we inspected correlations for total APRT scores by delay [49]. Specifically, we hypothesized that the self-report measures on the top half of the table (e.g., risk taking) were likely to relate to APRT performance, whereas those on the bottom half of the table (e.g., risk perception, externalizing, religiosity) were not. Because we obtained stronger effects for injury-related APRT factors than for reward-related factors, we expected stronger convergent validity for variables on the left side of the table than on the right side of the table. Thus, each APRT score (which constitute the table columns) had two families of correlations for convergent and divergent validity.

## Results

### Generalized estimating equations

The results of the GEEs are summarized in Table 2. Discernible main effects and interactions are explored in detail below. Table 3 summarizes the relevant means, standard errors, and Wald $\chi^2$ values for all statistically discernible interactions. A *post hoc* analysis revealed a correlation between reliability coefficient (Cronbach's α) and Wald $\chi^2$ (i.e., the effect size) values for the within-subject factors across all four APRT scores, $r(14) = .533$, $p = .034$.

**Points.** There was a main effect for Picture Type. The Hero condition resulted in the largest number of Points earned on average ($M = 114.23$, $SE = 5.42$), which was discernibly greater than the other three conditions. The Animal condition resulted in the lowest number of Points on average ($M = 69.09$, $SE = 5.42$), which was discernibly lower than all the other conditions. The average Points earned during Cliff ($M = 95.28$, $SE = 5.11$) and Disaster ($M = 100.94$, $SE = 5.70$) trials did not discernibly differ.

Participants earned more Points in low Injury Magnitude trials ($M = 143.53$, $SE = 7.62$) than in high Injury Magnitude trials ($M = 60.70$, $SE = 2.84$) and in low Injury Probability trials ($M = 169.19$, $SE = 8.24$) than in high Injury Probability trials ($M = 69.64$, $SE = 3.28$). They also earned more Points in high Reward Magnitude trials ($M = 292.98$, $SE = 13.34$) than in low Reward Magnitude trials ($M = 29.74$, $SE = 1.44$) and in high Reward Probability trials ($M = 124.80$, $SE = 5.66$) than in low Reward Probability trials ($M = 69.08$, $SE = 3.47$). After correcting for multiple comparisons, there was no discernible main effect of Delay on Points.

There was a Picture Type x Injury Probability interaction. There was no difference in average Points between Disaster and Hero trials when Injury Probability was low. When Injury Probability was high, Hero trials resulted in higher Points scores on average compared to Disaster trials. Also, the increase in Points from the Animal to the Cliff trials was slightly sharper when Injury Probability was high compared to when it was low.

There was an Injury Magnitude x Reward Magnitude interaction. When Reward Magnitude was high, the decrease in average Points per trial from the low Injury Magnitude to the high Injury Magnitude condition was sharper than when Reward Magnitude was low.

**Table 2. Summary of generalized estimating equation model effects (Wald $\chi^2$) for Points, Go Presses, Injuries, and Health.**

| Model Effect | df | Points | Go Presses | Remaining Health | Injuries |
|---|---|---|---|---|---|
| Picture Type | 3 | 45.60** | 87.07** | 109.74** | 77.94** |
| Injury Magnitude | 1 | 323.89** | 533.05** | 579.21** | 339.64** |
| Reward Magnitude | 1 | 4879.81** | 17.95** | 11.21** | 11.50** |
| Injury Probability | 1 | 863.75** | 1172.02** | 231.73** | 254.92** |
| Reward Probability | 1 | 258.11** | 2.75 | 8.51** | 2.69 |
| Delay | 1 | 3.59 | 9.00** | 27.75** | 22.13** |
| Picture Type x Injury Magnitude | 3 | 3.94 | 22.38** | 76.59** | 23.39** |
| Picture Type x Reward Magnitude | 3 | 2.26 | 3.98 | 21.84** | 7.68 |
| Picture Type x Injury Probability | 3 | 14.65** | 31.91** | 17.13** | 33.02** |
| Picture Type x Reward Probability | 3 | 1.05 | 5.80 | 11.96* | 27.72** |
| Injury Magnitude x Reward Magnitude | 1 | 24.27** | 0.30 | 0.04 | 6.14* |
| Injury Magnitude x Injury Probability | 1 | 34.07** | 71.19** | 9.44** | 1.46 |
| Injury Magnitude x Reward Probability | 1 | 3.51 | 1.91 | 0.38 | 0.001 |
| Reward Magnitude x Injury Probability | 1 | 10.98** | 2.59 | 1.14 | 0.02 |
| Reward Magnitude x Reward Probability | 1 | 0.90 | 14.87** | 1.56 | 16.00** |
| Injury Probability x Reward Probability | 1 | 15.74** | 0.06 | 0.01 | 0.01 |
| Picture Type x Delay | 3 | 4.62 | 8.62* | 17.08** | 3.54 |
| Injury Magnitude x Delay | 1 | 5.76* | 4.66* | 0.01 | 0.45 |
| Reward Magnitude x Delay | 1 | 0.03 | 0.17 | 0.05 | 0.03 |
| Injury Probability x Delay | 1 | 5.00* | 3.15 | 0.97 | 11.02** |
| Reward Probability x Delay | 1 | 0.48 | 0.03 | 5.74* | 0.07 |

*df* = degrees of freedom.

* $p < .05$ (uncorrected).

** $p_{adj} < .05$ (Holm-Sidak corrected).

There was an Injury Magnitude x Injury Probability interaction. When Injury Probability was low, the decrease in average Points going from the low Injury Magnitude condition to the high Injury Magnitude condition was slightly sharper than when Injury Probability was high.

There was a Reward Magnitude x Injury Probability interaction. When Reward Magnitude was high, the decrease in average Points going from the low Injury Probability condition to the high Injury Probability condition was sharper compared to when it was low.

Finally, there was an Injury Probability x Reward Probability interaction. When Injury Probability was low, the increase in average Points going from the low Reward Probability condition to the high Reward Probability condition was slightly sharper compared to when it was high. No other discernible within-subject interactions for Points were observed. After correcting for multiple comparisons, no discernible interactions involving Delay were observed.

**Go Presses.** There was a main effect for Picture Type. All picture categories had discernibly different Go Presses from each other. The Hero condition had the most Go Presses on average ($M = 5.98$, $SE = 0.27$), followed by Disaster ($M = 5.31$, $SE = 0.27$), Cliff ($M = 4.85$, $SE = 0.24$), and Animal ($M = 4.05$, $SE = 0.26$).

Participants made more Go Presses in low Injury Magnitude trials ($M = 6.55$, $SE = 0.35$) than in high Injury Magnitude trials ($M = 3.81$, $SE = 0.17$) and in low Injury Probability trials ($M = 7.36$, $SE = 0.38$) than in high Injury Probability trials ($M = 3.40$, $SE = 0.16$). Participants also made more Go Presses in low Reward Magnitude trials ($M = 5.23$, $SE = 0.25$) than in high Reward Magnitude trials ($M = 4.78$, $SE = 0.24$). Finally, participants who were given the forced

**Table 3. Decompositions of statistically discernible APRT interactions.**

| | Low v1/Low v2 | | Low v1/High v2 | | | High v1/Low v2 | | High v1/High v2 | | |
|---|---|---|---|---|---|---|---|---|---|---|
| Interaction (v1/v2) | M | SE | M | SE | Wald $\chi^2$(1) | M | SE | M | SE | Wald $\chi^2$(1) |
| | | | | | *POINTS* | | | | | |
| Injury Probability/Disaster-Hero | 196.94 | 10.80 | 196.35 | 10.26 | 0.00 | 51.74 | 3.87 | 66.45 | 3.83 | 8.70* |
| Injury Probability/Animal-Cliff | 131.28 | 10.20 | 161.43 | 9.24 | 9.00* | 36.36 | 3.55 | 56.24 | 3.88 | 18.16*** |
| Injury Magnitude/Reward Magnitude | 49.06 | 2.76 | 419.91 | 22.81 | 326.62*** | 18.02 | 1.11 | 204.41 | 9.16 | 467.55*** |
| Injury Magnitude/Injury Probability | 236.00 | 13.6 | 87.29 | 4.68 | 215.96*** | 121.29 | 5.586 | 30.38 | 1.98 | 273.27*** |
| Reward Magnitude/Injury Probability | 56.34 | 2.95 | 15.69 | 0.92 | 242.57*** | 508.05 | 26.58 | 168.95 | 7.94 | 238.68*** |
| Injury Probability/Reward Probability | 133.38 | 6.76 | 214.62 | 11.12 | 138.29*** | 36.54 | 2.29 | 72.57 | 3.42 | 150.27*** |
| | | | | | *GO PRESSES* | | | | | |
| Injury Magnitude/Disaster-Hero | 7.38 | 0.40 | 7.46 | 0.39 | 0.15 | 3.82 | 0.21 | 4.80 | 0.21 | 13.80*** |
| Injury Magnitude/Animal-Cliff | 5.47 | 0.36 | 6.12 | 0.36 | 10.46** | 3.00 | 0.21 | 3.84 | 0.19 | 21.78*** |
| Injury Probability/Disaster-Hero | 8.48 | 0.44 | 8.54 | 0.43 | 0.05 | 3.33 | 0.19 | 4.19 | 0.19 | 29.57*** |
| Injury Probability/Animal-Cliff | 5.90 | 0.42 | 6.85 | 0.38 | 11.51** | 2.78 | 0.18 | 3.43 | 0.18 | 21.56*** |
| Injury Magnitude/Injury Probability | 9.02 | 0.51 | 4.79 | 0.25 | 202.59*** | 6.00 | 0.30 | 2.43 | 0.11 | 264.80*** |
| Reward Magnitude/Reward Probability | 5.50 | 0.27 | 4.96 | 0.25 | 14.88*** | 4.69 | 0.25 | 4.88 | 0.25 | 2.14 |
| | | | | | *REMAINING HEALTH* | | | | | |
| Injury Magnitude/Disaster-Hero | 77.99 | 1.20 | 78.03 | 1.19 | 0.003 | 62.56 | 1.59 | 56.03 | 1.47 | 35.24*** |
| Injury Magnitude/Animal-Cliff | 82.39 | 1.11 | 81.61 | 1.11 | 1.40 | 70.43 | 1.54 | 64.91 | 1.44 | 23.45*** |
| Reward Magnitude/Disaster-Hero | 67.22 | 1.52 | 68.02 | 1.38 | 0.35 | 73.32 | 1.36 | 66.04 | 1.35 | 51.08*** |
| Reward Magnitude/Animal-Cliff | 75.19 | 1.45 | 73.28 | 1.31 | 2.15 | 77.64 | 1.34 | 73.25 | 1.34 | 12.74** |
| Injury Probability/Disaster-Hero | 72.76 | 1.36 | 72.17 | 1.31 | 0.36 | 67.79 | 1.44 | 61.88 | 1.37 | 29.30*** |
| Injury Probability/Animal-Cliff | 81.04 | 1.24 | 77.23 | 1.23 | 14.52*** | 71.79 | 1.42 | 69.29 | 1.35 | 5.84 |
| Injury Magnitude/Injury Probability | 83.49 | 1.04 | 76.52 | 1.45 | 186.78*** | 68.11 | 1.32 | 58.85 | 1.45 | 148.13*** |
| Delay/Disaster-Hero | 64.11 | 2.00 | 61.25 | 2.00 | 5.87 | 64.11 | 1.65 | 72.80 | 1.53 | 9.05*** |
| Delay/Animal-Cliff | 71.94 | 2.00 | 65.15 | 2.12 | 28.14** | 80.88 | 1.56 | 81.37 | 1.8 | 0.18 |
| | | | | | *INJURIES* | | | | | |
| Injury Magnitude/Disaster-Hero | 1.00 | 0.05 | 1.03 | 0.05 | 1.00 | 0.51 | 0.02 | 0.63 | 0.02 | 30.25*** |
| Injury Magnitude/Animal-Cliff | 0.77 | 0.05 | 0.86 | 0.05 | 7.11** | 0.40 | 0.03 | 0.50 | 0.02 | 20.25*** |
| | Low v1/Low v2 | | Low v1/High v2 | | | High v1/Low v2 | | High v1/High v2 | | |
| Interaction (v1/v2) | M | SE | M | SE | Wald $\chi^2$(1) | M | SE | M | SE | Wald $\chi^2$(1) |
| | *INJURIES* | | | | | | | | | |
| Injury Probability/Disaster-Hero | 0.65 | 0.03 | 0.65 | 0.03 | 0.004 | 0.79 | 0.04 | 1.00 | 0.04 | 29.30*** |
| Injury Probability/Animal-Cliff | 0.46 | 0.03 | 0.52 | 0.03 | 9.00 | 0.68 | 0.04 | 0.82 | 0.04 | 21.78*** |
| Reward Probability/Disaster-Hero | 0.74 | 0.04 | 0.77 | 0.03 | 16.00*** | 0.70 | 0.04 | 0.84 | 0.04 | 21.78*** |
| Reward Probability/Animal-Cliff | 0.60 | 0.04 | 0.65 | 0.03 | 4.00 | 0.52 | 0.03 | 0.66 | 0.03 | 49.00*** |
| Reward Magnitude/Reward Probability | 0.74 | 0.03 | 0.67 | 0.03 | 16.89*** | 0.64 | 0.03 | 0.66 | 0.03 | 1.67*** |
| Delay/Injury Probability | 0.75 | 0.05 | 0.96 | 0.06 | 144.00*** | 0.44 | 0.03 | 0.70 | 0.04 | 169.00*** |

For interactions involving Delay, "Low" denotes non-delayed participants, and "High" denotes delayed participants. For interactions involving Animal/Cliff/Hero/Disaster, Animal and Disaster are considered "Low" and Cliff and Hero are considered "High."

* $p_{adj}$ < .05.

** $p_{adj}$ < .01.

*** $p_{adj}$ < .001.

All $p$ values reflect a Holm-Sidak correction for multiple comparisons.

delay between responses had fewer Go Presses ($M$ = 4.35, $SE$ = 0.29) on average than participants that were not delayed ($M$ = 5.75, $SE$ = 0.39). After correcting for multiple comparisons, there was no main effect for Reward Probability.

There was a Picture Type x Injury Magnitude interaction. There was no discernible difference in Go Presses between Disaster and Hero trials when Injury Magnitude was low, but when Injury Magnitude was high, Hero trials resulted in discernibly more Go Presses compared to Disaster trials. Additionally, when Injury Magnitude was high, the increase in Go Presses when going from Animal trials to Cliff trials was slightly sharper compared to when it was low.

There was a Picture Type x Injury Probability interaction. When Injury Probability was low, there was no difference in Go Presses between Disaster and Hero trials, but Hero trials resulted in discernibly more Go Presses compared to Disaster trials when Injury Probability was high. Additionally, when Injury Probability was low, the increase in Go Presses going from Animal to Cliff trials was slightly sharper compared to when it was high.

There was an Injury Magnitude x Injury Probability interaction. When Injury Probability was low, the decrease in Go Presses going from the low Injury Magnitude to the high Injury Magnitude condition was slightly sharper compared to when it was high.

Finally, there was a Reward Magnitude x Reward Probability interaction. When Reward Probability was high, there was no difference in Go Presses between the low and high Reward Magnitude conditions, but high Reward Magnitude trials resulted in discernibly fewer Go Presses compared to low Reward Magnitude trials when Reward Probability was low. After correcting for multiple comparisons, no discernible interactions involving Delay were observed for Go Presses.

**Remaining Health.**   There was a main effect for Picture Type. All picture categories had discernibly different amounts of Remaining Heath from each other on average. Animal trials had the highest Remaining Health on average ($M = 76.41$, $SE = 1.26$), followed by Cliff trials ($M = 73.26$, $SE = 1.19$), Disaster trials ($M = 70.27$, $SE = 1.31$), and Hero trials ($M = 67.02$, $SE = 1.25$).

Participants had higher Remaining Health on average in low Injury Magnitude trials ($M = 80$, $SE = 1.06$) than in high Injury Magnitude trials ($M = 63.48$, $SE = 1.33$) and in low Injury Probability trials ($M = 75.80$, $SE = 1.13$) than in high Injury Probability trials ($M = 67.69$, $SE = 1.24$). Participants also had greater Remaining Health in high Reward Magnitude trials ($M = 72.56$, $SE = 1.16$) than in low Reward Magnitude trials ($M = 70.92$, $SE = 1.20$) and slightly more Remaining Health in high Reward Probability trials on average ($M = 72.34$, $SE = 1.17$) than in low Reward Probability trials ($M = 71.15$, $SE = 1.18$). Finally, participants who were given the delay had higher Remaining Health on average ($M = 77.87$, $SE = 1.43$) than participants not given the delay ($M = 65.61$, $SE = 1.82$).

There was a Picture Type x Injury Magnitude interaction. When Injury Magnitude was low, there was no discernible difference in average Remaining Health between Disaster and Hero trials, but Hero trials resulted in discernibly lower Remaining Health scores compared to Disaster trials when Injury Magnitude was high. Similarly, for Animal and Cliff trials, there was no discernible difference in Health scores between the two when Injury Magnitude was low; but when Injury Magnitude was high, Cliff trials resulted in lower Remaining Health on average compared to Animal trials.

There was a Picture Type x Reward Magnitude interaction. When Reward Magnitude was low, there was no discernible difference in Remaining Health scores between Disaster and Hero trials, but when Reward Magnitude was high, Hero trials resulted in discernibly lower Remaining Health scores on average compared to Disaster trials. A similar effect was observed for Animal and Cliff trials, wherein there was no discernible difference between them when Reward Magnitude was low, but when Reward Magnitude was high, Cliff trials resulted in discernibly lower Health scores on average compared to Animal trials.

There was a Picture Type x Injury Probability interaction. When Injury Probability was low, there was no discernible difference in Health scores between Disaster and Hero trials. When Injury Probability was high, Hero trials resulted in discernibly lower Health scores on average compared to Disaster trials. In contrast, when Injury Probability was low, Cliff trials resulted in discernibly lower Health scores compared to Animal trials, but there was no discernible difference in scores between Cliff and Animal trials when Injury Probability was high.

There was an Injury Magnitude x Injury Probability interaction. When Injury Magnitude was high, the decrease in Remaining Health score when going from the low Injury Probability condition to the high Injury Probability condition was slightly sharper compared to when Injury Magnitude was low.

Finally, there was a Picture Type x Delay interaction. For non-delayed participants, Hero trial Remaining Health scores were not discernibly different from Disaster trial scores, but Hero trials resulted in discernibly higher Health scores compared to Disaster trials in the delayed participants.

**Injuries.** There was a main effect for Picture Type. Each picture category differed discernibly from the other three in average Injuries for each trial. Participants had discernibly greater Injuries in low Injury Magnitude trials ($M = 0.91$, $SE = 0.05$) than in high Injury Magnitude trials ($M = 0.50$, $SE = 0.02$) and in high Injury Probability trials ($M = 0.82$, $SE = 0.03$) than in low Injury Probability trials ($M = 0.56$, $SE = 0.03$). Participants suffered discernibly more Injuries in low Reward Magnitude trials ($M = 0.70$, $SE = 0.03$) than in high Reward Magnitude trials ($M = 0.65$, $SE = 0.03$). Finally, participants that were given the delay had discernibly fewer Injuries on average ($M = 0.55$, $SE = 0.03$) than participants that were not given the delay ($M = 0.83$, $SE = 0.05$). After controlling for multiple comparisons, there was no main effect for Reward Probability.

There was a Picture Type x Injury Magnitude interaction. When Injury Magnitude was low, there was no discernible difference in average Injuries per trial between Disaster and Hero trials, but there were discernibly more Injuries during Hero trials compared to Disaster trials when Injury Magnitude was high. Additionally, the increase in average Injuries score when going from Animal to Cliff trials was slightly sharper when Injury Magnitude was high compared to when it was low.

There was a Picture Type x Injury Probability interaction. When Injury Probability was low, there was no discernible difference in Injury score between Disaster and Hero trials, but Hero trials resulted in discernibly more average Injuries compared to Disaster trials when Injury Probability was high. Additionally, when Injury Probability was low, there was no discernible difference in Injuries score between Animal and Cliff trials, but Cliff trials resulted in discernibly more Injuries on average compared to Animal trials when Injury Probability was high.

There was a Picture Type x Reward Probability interaction. When Reward Probability was high, the increase in average Injuries per trial when going from Disaster to Hero trials was much sharper compared to when Reward Probability was low. Additionally, the increase in Injuries score when going from Animal to Cliff trials was also much sharper when Reward Probability was high compared to when it was low.

There was a Reward Magnitude x Reward Probability interaction. When Reward Probability was high, there was no discernible difference in Injuries score between the low and high Reward Magnitude conditions, but high Reward Magnitude trials resulted in discernibly fewer injuries compared to low Reward Magnitude trials when reward Probability was high.

Finally, there was an Injury Probability x Delay interaction. The increase in average Injuries score when going from low Injury Probability to high Injury Probability was sharper in the delayed participants compared to the non-delayed participants.

## Correlations

Because the patterns of correlations within non-delay and delay groups were discernibly different for all four APRT scores, $\chi^2(42)$s range from 87.8–520, $ps < .00001$, Tables 4 and 5 display the correlations between the APRT Go Press and Points scores (respectively) and the self-

**Table 4. Pearson correlations between APRT Go Presses scores and self-report measures by delay condition.**

| Self-Report Scale | Injury Magnitude | Injury Probability | Animal-Cliff | Hero-Disaster | Reward Magnitude | Reward Probability | APRT total |
|---|---|---|---|---|---|---|---|
| | **APRT SCORE (NO DELAY/DELAY)** | | | | | | |
| SSS Thrill-Seeking | -.253*/-.119 | -.343***/-.152 | .073/.052 | .094/.181 | -.035/-.085 | -.059/-.076 | .322***/.208 |
| DOSPERT Recreational Risk taking | -.297**/-.097 | -.349***/-.182 | -.049/-.047 | .074/.103 | -.077/-.120 | -.074/-.026 | .331***/.240 |
| UPPS-P Sensation Seeking | -.295**/-.093 | -.335***/-.193 | .028/.035 | .097/.142 | -.079/-.031 | -.116/-.000 | .323**/.194 |
| SHART Driving Risk taking | -.167/-.151 | -.192/-.182 | .123/-.037 | -.127/.095 | .016/-.005 | -.045/.092 | .202/.180 |
| DOSPERT Social Risk taking | -.085/-.228 | -.135/-.024 | -.007/-.022 | .128/-.091 | .103/.008 | .012/-.037 | .132/.229 |
| DOSPERT Health/Safety Risk taking | .002/-.245* | -.062/-.160 | .084/.022 | -.008/.036 | -.084/.112 | -.114/.073 | .075/.173 |
| DOSPERT Ethics Risk taking | .020/-.050 | -.129/-.070 | -.009/.186 | -.066/-.075 | -.011/-.009 | -.336***/.221 | .130/.104 |
| SHART Sexual Risk taking | .075/.016 | -.023/-.117 | .152/-.051 | -.080/-.056 | -.066/-.024 | -.007/.173 | .036/.053 |
| DOSPERT Financial Risk taking | -.076/.054 | -.057/-.046 | .006/.062 | .026/.053 | -.056/-.021 | -.070/.087 | .042/-.029 |
| DOSPERT Recreational Risk-Perception | .253*/.115 | .249*/.140 | .089/-.059 | -.103/-.224 | .024/.008 | .086/.011 | -.252*/-.167 |
| SHART Driving Risk-Perception | .116/.197 | .147/.214 | .078/-.061 | .111/-.079 | -.004/-.014 | -.024/-.058 | -.172/-.204 |
| BHRQ Adulthood Aggressive | -.085/-.148 | -.085/-.087 | .015/-.154 | -.024/-.010 | .029/-.016 | -.100/.092 | .123/.129 |
| SSS Boredom Susceptibility | -.140/-.071 | -.207/.003 | .030/-.051 | .018/-.058 | .063/-.044 | -.144/-.072 | .258*/.066 |
| SHART Sexual Risk-Perception | .182/.042 | .227/.063 | -.046/-.086 | .051/.065 | -.173/-.079 | -.004/-.047 | -.238/-.051 |
| DOSPERT Ethical Risk-Perception | .102/.103 | .206/.041 | .053/-.261* | .016/-.076 | -.071/.006 | .233/-.101 | -.201/-.048 |
| BHRQ Childhood Aggressive | .085/-.139 | .066/-.139 | .018/-.088 | -.089/-.071 | -.028/-.152 | -.111/.156 | -.027/.141 |
| DOSPERT Health/Safety Risk-Perception | .068/.209 | .102/.095 | .043/-.153 | -.081/-.127 | -.052/.048 | .050/-.042 | -.127/-.095 |
| ADS Total | -.041/-.033 | -.118/-.101 | .039/-.108 | -.085/.004 | -.067/.009 | -.056/.151 | .141/.027 |
| BHRQ Childhood Non-aggressive | .103/-.045 | .023/-.130 | -.003/-.134 | -.034/-.088 | .017/-.241 | -.019/.170 | -.016/.124 |
| DOSPERT Financial Risk-Perception | .132/.076 | .228/.029 | -.011/-.188 | .046/.036 | .079/.062 | .037/.075 | -.192/-.087 |
| SDAST Total | .101/-.017 | .055/-.170 | -.014/-.117 | .066/.053 | -0.199/.061 | -.118/.133 | -.003/.112 |
| SSS Experience Seeking | .020/-.033 | .016/-.027 | .062/.066 | .114/-.009 | .070/.063 | .053/.023 | .040/.008 |
| UPPS-P Negative Urgency | .089/.078 | .042/-.022 | .175/.014 | -.081/.014 | -.008/-.126 | .002/.139 | .001/.038 |
| CRS Total | .083/-.051 | .103/-.065 | .058/-.189 | .051/.054 | -.033/-.194 | .074/-.042 | -.066/.035 |
| UPPS-P Perseverance | .179/.126 | .141/.068 | .106/.010 | -.001/-.042 | -.026/-.081 | -.050/.081 | -.108/-.079 |
| UPPS-P Positive Urgency | .074/.114 | -.022/.051 | .037/-.018 | .019/-.004 | -.073/-.144 | -.164/.058 | .029/-.038 |
| SSS Disinhibition | -.008/.097 | -.051/.048 | .009/.005 | -.088/-.021 | -.046/.132 | .056/.056 | .088/-.070 |
| BHRQ Adulthood Non-aggressive | .050/-.019 | -.018/-.043 | .094/-.086 | .002/.063 | .021/-.011 | -.081/.195 | .023/-.013 |
| UPPS-P Lack of Premeditation | .061/-.152 | -.004/-.063 | .199*/.016 | -.045/-.074 | .142/-.027 | .034/.046 | .021/.056 |
| DOSPERT Social Risk-Perception | .055/.019 | .061/-.044 | -.035/-.116 | -.123/.016 | -.049/-.027 | .050/.015 | -.043/-.003 |

SSS = Sensation Seeking Scale, DOSPERT = Domain Specific Risk Taking Scales, SHART = Sexual, Health, and Automotive Risk-Taking Scales, BHRQ = Behavioral Report on Rule-Breaking Questionnaire, ADS = Alcohol Dependency Scale, SDAST = Short Drug Abuse Screening Test, CRS = Centrality of Religion Scale.

* $p_{adj} < .05$.

** $p_{adj} < .01$.

*** $p_{adj} < .001$.

All $p$ values reflect a Holm-Sidak correction for multiple comparisons.

**Table 5. Pearson correlations between APRT Points scores and self-report measures by delay condition.**

| Self-Report Scale | APRT SCORE (NO DELAY/DELAY) | | | | | | |
|---|---|---|---|---|---|---|---|
| | Injury Magnitude | Injury Probability | Animal-Cliff | Hero-Disaster | Reward Magnitude | Reward Probability | APRT total |
| SSS Thrill-Seeking | -.310***/-.172 | -.324***/-.128 | .040/-.042 | .058/.121 | .343***/.167 | .198/.089 | .355***/.179 |
| DOSPERT Recreational Risk taking | -.353***/-.125 | -.318***/-.174 | -.078/-.106 | .122/.113 | .322***/.198* | .233/.147 | .342***/.209 |
| UPPS-P Sensation Seeking | -.336***/-.135 | -.280**/-.200 | -.026/-.024 | .123/.095 | .295**/.197 | .198/.155 | .315***/.202 |
| SHART Driving Risk taking | -.202/-.193 | -.235/-.182 | .055/-.044 | -.094/.162 | .248*/.180 | .167/.227 | .252*/.180 |
| DOSPERT Social Risk taking | -.166/-.264* | -.214/-.189 | -.118/.085 | .070/-.012 | .233/.208 | .324***/.123 | .221/.215 |
| DOSPERT Health/Safety Risk taking | -.030/-.225 | -.102/-.188 | .047/.050 | .054/.183 | .100/.170 | .015/.135 | .111/.167 |
| DOSPERT Ethics Risk taking | -.081/-.070 | -.116/-.102 | -.074/.111 | .026/.006 | .151/.099 | .071/.193 | .148/.095 |
| SHART Sexual Risk taking | .091/-.005 | -.080/-.090 | .115/-.117 | -.035/-.020 | .036/.067 | .024/.169 | .043/.062 |
| DOSPERT Financial Risk taking | -.057/.077 | -.043/-.028 | .020/.127 | .104/.055 | .035/-.050 | -.003/.044 | .043/-.042 |
| DOSPERT Recreational Risk-Perception | .292**/.168 | .231/.101 | .060/.011 | -.153/-.204 | -.243/-.171 | -.131/-.069 | -.254*/-.170 |
| SHART Driving Risk-Perception | .177/.227 | .146/.206 | .143/.008 | .102/-.091 | -.185/-.205 | -.099/-.153 | -.181/-.211 |
| BHRQ Adulthood Aggressive | -.126/-.169 | -.131/-.145 | -.007/-.186 | .071/-.001 | .160/.147 | .146/.164 | .158/.140 |
| SSS Boredom Susceptibility | -.224/-.091 | -.206/-.033 | -.147/.032 | .030/.048 | .258*/.045 | .228/-.032 | .252*/.050 |
| SHART Sexual Risk-Perception | .200/.091 | .239/.068 | .036/-.064 | .062/-.049 | -.229/-.086 | -.086/-.075 | -.216/-.075 |
| DOSPERT Ethical Risk-Perception | .196/.086 | .251*/.025 | .079/-.178 | -.044/-.109 | -.259*/-.050 | -.136/-.075 | -.251*/-.047 |
| BHRQ Childhood Aggressive | .066/-.117 | .016/-.144 | -.010/-.134 | .068/.010 | -.009/.107 | -.044/.188 | -.012/.117 |
| DOSPERT Health/Safety Risk-Perception | .152/.182 | .129/.083 | .048/-.053 | -.169/-.126 | -.144/-.096 | .002/-.079 | -.132/-.097 |
| ADS Total | -.079/-.027 | -.171/-.085 | -.003/-.141 | .021/-.009 | .154/.026 | .013/.120 | .159/.024 |
| BHRQ Childhood Non-aggressive | .121/-.117 | .001/-.098 | -.010/-.228 | .079/-.097 | .001/.074 | .058/.177 | -.004/.088 |
| DOSPERT Financial Risk-Perception | .102/.072 | .219/.018 | .015/-.249* | -.058/-.125 | -.132/-.062 | -.155/.053 | -.141/-.070 |
| SDAST Total | .054/-.011 | .068/-.157 | .051/-.213 | .087/-.024 | -.021/.136 | -.089/.152 | -.006/.125 |
| SSS Experience Seeking | -.074/.015 | -.066/-.070 | -.004/.083 | .115/-.026 | .097/.011 | -.019/.014 | .084/.009 |
| UPPS-P Negative Urgency | .088/.103 | .063/-.034 | .166/-.045 | -.079/.045 | -.020/.029 | -.050/.107 | -.020/.037 |
| CRS Total | .141/-.011 | .140/.014 | .130/-.239 | .045/-.072 | -.123/.001 | -.063/.046 | -.116/.014 |
| UPPS-P Perseverance | .151/.155 | .128/.092 | .111/.072 | -.020/.061 | -.134/-.100 | .000/-.065 | -.133/-.089 |
| BHRQ Adulthood Non-aggressive | .050/.139 | .031/.026 | .003/-.003 | .067/.133 | -.011/-.057 | -.048/.011 | -.002/-.046 |
| UPPS-P Lack of Premeditation | -.076/.088 | -.111/-.032 | -.007/-.031 | .007/-.003 | .122/-.047 | .058/-.017 | .122/-.060 |
| DOSPERT Social Risk-Perception | .019/-.013 | -.068/-.050 | .045/-.126 | .064/-.013 | .051/-.011 | .092/.107 | .051/-.014 |
| SSS Thrill-Seeking | .050/-.109 | .004/-.080 | .131/.119 | -.075/.166 | .038/.024 | .029/.038 | .023/.038 |
| DOSPERT Recreational Risk taking | .073/.049 | .051/-.061 | .052/-.065 | -.035/.046 | -.063/.005 | -.123/.036 | -.062/.011 |

SSS = Sensation Seeking Scale, DOSPERT = Domain Specific Risk Taking Scales, SHART = Sexual, Health, and Automotive Risk-Taking Scales, BHRQ = Behavioral Report on Rule-Breaking Questionnaire, ADS = Alcohol Dependency Scale, SDAST = Short Drug Abuse Screening Test, CRS = Centrality of Religion Scale.

* $p_{adj} < .05$.

** $p_{adj} < .01$.

*** $p_{adj} < .001$.

All $p$ values reflect a Holm-Sidak correction for multiple comparisons.

report measures of risk taking and non-risk constructs, separated by Delay condition. Because the pattern of correlations was very similar for Go Presses, Remaining Health, and Injuries scores, the correlation tables for the latter two indirect measures of risk taking are provided in

S4 and S5 Tables, respectively. The vertical lines distinguish the scales we believed assessed risk taking from the scales we believed assessed other constructs (or at least were weaker measures of risk taking), with the horizontal line also marking where a new family of correlations began in a given column within each table. Both axes of the tables were ordered such that the strongest measures of risk taking will be closest to the top/left side of the table. The closer a measure was listed to the bottom/right, the weaker a measure of risk taking we hypothesized it to be.

Overall, APRT scores (both difference scores and total score) correlated most strongly with self-report measures of thrill seeking (i.e., SSS Thrill and Adventure Seeking, UPPS-P Sensation Seeking) and physical risk taking (i.e., DOSPERT Recreational risk taking). The injury-related difference scores and total scores had the greatest number of discernible correlations, especially for Go Presses and Points. At the same time, APRT scores universally exhibited weak to no discernible relationships with the non-risk taking measures (e.g., CRS, SSS Disinhibition) and measures of risk perception (e.g., DOSPERT risk perception scales), providing good evidence of the measure's divergent validity.

Many correlations that were discernible in the non-delayed participants were not discernible in delayed participants. This was true across all four sets of APRT difference and total scores. Though there were some exceptions, the most pertinent correlations (i.e., those involving self-report measures of physical risk or adventure/thrill-seeking) exhibited this pattern.

## Discussion

This study assessed how specific aspects of risk influenced scores on the novel Assessment of Physical Risk Taking, examined the convergent and divergent validity of this measure, and determined whether delaying consecutive responses would substantially alter any correlations the measure scores have with self-report measures of risk taking.

### Effects of injury, reward magnitude, probability, and situation

**Within-task main effects.** Regarding the first goal of this study, there were discernible and strong main effects for nearly all the within-subject variables across all four APRT scores. Injury Magnitude and Probability affected participants' behavior and scores more strongly than Reward Magnitude and Probability when making risky decisions. This is especially striking when comparing Reward Probability (which had few discernible effects and weak effect sizes for those that were significant) to Injury Probability (which had the largest number of main effects/interactions and some of the strongest effect sizes in all four GEE models). Picture Type fell somewhere between these two variables. Because the sizes of the effects and their internal consistencies had a large relationship, changes to APRT that improve the reliability of scores may reveal stronger effects for many of the variables that exhibited low Wald $\chi2$ values in the present study.

Compared to other behavioral measures of risk taking, APRT illustrates the importance of injury variables even more directly than other tasks. Whereas BART has three injury probability conditions (compared to APRT's two), there is usually not enough individual variation in the two riskiest conditions to warrant including them in analyses [15]. As a result, many authors use only the lowest injury probability category [50]. BART also lacks any variation in injury magnitude, as the only negative outcome is the popping of the balloon, which is analogous to APRT's "death" screen, rather than APRT's varying Injury Magnitude between trials. The IGT provides a more variegated assessment of injury magnitude, as the primary outcome score is the difference in number of card choices from high-risk (i.e., higher injury magnitude and probability) vs. low-risk decks [20]. The current results from APRT are consistent with similar findings on IGT performance, wherein higher injury magnitude/probability reduces

risky behavior (e.g., fewer button presses or less risky card choices) and is weighted more heavily than reward magnitude/probability [28, 51]. Similarly, the CCT explicitly manipulates both the injury magnitude and probability across its trials, making it the best current risk taking measure of the effects of these variables [24]. For these variables, the CCT has shown effects similar to the APRT, wherein increased risk decreases the number of cards most participants will choose before ending a trial, and the effects of increased loss probability/magnitude outweigh the effects of gain probability/magnitude [26, 28].

The difference in Animal and Cliff trial scores tended to increase when either Injury Magnitude or Injury Probability was high, suggesting increased defensive emotional processing when faced with a dangerous animal. This behavioral response is consistent with pictures of threatening animals serving as potent potentiators of the defensive startle blink reflex [52, 53]. Similarly, the difference in Hero and Disaster trial scores was only discernibly different when Injury Probability or Magnitude was high, with all APRT scores being higher (except Health, which is expected from receiving more injuries) in the Hero condition. These Hero-Disaster interactions suggest that participants were more willing to risk harm to self when the stakes were higher, but only if the risk was related to attempting to help another person in danger. This is consistent with some of the literature on risk taking and altruism, which posits that individuals are more willing to accept higher risks if it is for the sake of aiding another person [54, 55].

**Interactions and the meaning of physical risk taking.** Overall, there were more discernible interactions involving Injury Magnitude and Probability than Reward Magnitude and Probability (as shown in Table 3), and the majority of the Injury-related interactions were larger in effect size than the majority of the Reward-related interactions. Again, this pattern is consistent with findings from other risk taking measures. The loss probability and amount variables within the CCT have been shown to have stronger interactions with other variables (such as personality) than their corresponding gain variables in the task, providing further support for the construct validity of APRT's variables [27, 56].

The overall pattern of results across each of the APRT scores seems to suggest that the task assesses risk taking in a manner different to that of other tasks, like the BART. Specifically, risk taking in APRT can be modeled as the magnitude of the difference in scores between the Injury variable conditions, rather than as an overall score. Higher risk taking on the APRT is the failure of a participant to alter their behavior (e.g., Go Presses) in response to information indicating an increased chance of Injury. That is, participants high in risk taking would evidence little to no difference in Go Presses when Injury Magnitude or Probability switches from low to high). In contrast, individuals with low or normal levels of risk taking would tend to heed this information more readily and would have fewer button presses in the high injury magnitude and probability conditions than in the low conditions.

This tendency to ignore increases in risk can be further understood as the opposite of a satisficing drive that seems to be present in the normal population. This is evidenced within APRT through the pattern of the Points scores, which indicated that most participants would become satisfied with the amount of points they have earned in a trial after reaching a certain limit. At that point, they would quit the trial, even if Injury Probability and/or Magnitude was low and Reward Magnitude was high (i.e., when they had low-risk opportunities to earn large amounts of Points indefinitely). Thus, APRT presents a novel conceptualization of risk taking: a contextual insensitivity to increases in risk. This definition complements the concept of satisficing, which describes the tendency of most individuals to pursue a suboptimal target (i.e., not maximizing reward) in exchange for decreased risk [57]. Satisficing has been more heavily examined (with respect to risk taking) in behavioral economics than in other branches of psychology [58]. APRT thus potentially merges risk taking research across multiple domains

under a uniform conceptualization of risky behavior by providing separate operationalizations of risk taking and satisficing (Go Presses and Points, respectively).

## Convergent and divergent validity of APRT scores

The pattern of correlations that emerged provided evidence for the construct validity of APRT as a more precise measure of physical risk taking than we had anticipated. Measures of sensation seeking, thrill-seeking, and physical risk (e.g., SSS–Thrill Seeking, DOSPERT Recreational risk taking) had the strongest correlations across all APRT scores, whereas the weakest correlations were the measures that were not measures of risk taking (e.g., CRS, BHRQ scales, DOSPERT risk perception scales). This pattern of convergent and divergent validity suggests the APRT assesses physical risk taking in particular, in contrast to other measures' associations with broader risk taking constructs. BART performance has repeatedly exhibited correlations of similar magnitude (i.e., $r$s = .20-.35) with broader measures of impulsivity and sensation seeking [15, 50, 59, 60]. Unlike APRT, however, BART performance also correlates with self-reported substance use, reckless driving, and risky sexual behavior ($r$s: .25-.28), though the replicability of these correlates has been questioned [15, 50]. Thus, though APRT seems to be more specific in its correlates than comparable measures, the strength of its relationships is consistent with what would be expected when comparing task performance with relevant personality traits.

Additionally, the difference scores for the Injury-related factors within APRT for each score were generally more strongly correlated with these physical risk taking questionnaires than Reward or Picture Type difference scores. This pattern further supports the notion that changes in injury probability or magnitude are more important factors when considering a risky decision than changes in reward magnitude/probability or the specific risky situation. However, the overall total for each APRT score had equal or even slightly stronger correlations with the convergent self-report measures than the Injury-related difference scores, suggesting overall task performance was associated with their self-reported physical risk taking tendency. Consequently, total scores on the APRT may be reasonable measures of risk taking to use in future studies.

Nevertheless, the Injury-related difference scores for Go Presses may best measure physical risk taking, as these scores most clearly follow APRT's conceptualization of physical risk taking. This is especially true when used in studies that use other measures of physical risk taking, which until this study has been defined simply as "engaging in a behavior that can lead to injury when there are alternatives that do not pose such risk of injury" [61]. Comparing the operational differences between these two conceptualizations will be critical for the future of the risk taking literature. Beyond physical risk taking, many other risk taking studies focus on sensitivity to punishment and reward, especially those examining differences in risk taking between adolescents and adults [62]. APRT's ability to examine insensitivity to changes in risk could prove invaluable to better understanding how risk taking propensity changes as an adolescent matures into adulthood [63]. On the other hand, APRT total Go Presses may be preferable to use in studies where a basic measure of an individual's physical risk taking behavior is desired. Both its stronger convergent validity with self-report measures of physical risk taking and its divergent validity with measures of other types of risk taking and risk-related personality traits recommend it for this purpose.

## Effects of delay: Hot versus cool risk taking

Overall, imposing a delay reduced participants' go presses and points in the APRT, leaving them with fewer injuries and more health at the end of trials. Such a "cool" response pattern

indicates that delayed participants either played the overall task more cautiously or were more bored during the APRT than non-delayed participants. The lack of interactions with Delay suggests that participants were not simply bored with the task, as Delay rarely moderated the effects of the other manipulations on scores. This bodes well for APRT's internal validity, as it indicates the "cool" and "hot" versions of the APRT measure similar constructs.

Indeed, the discernible differences in correlations between the "hot" and "cool" APRT versions imply that delayed participants were more cautious during the APRT than they reported themselves being in daily life. The majority of correlations across all APRT scores were much weaker (to the point of being indiscernible) in delayed participants than non-delayed participants. This pattern suggests that the delay at least partially negates differences in risk taking propensity between individuals, with the forced wait allowing ordinarily impulsive individuals enough time to reconsider their decisions and possibly restrain their behavior more frequently than when they can respond immediately [33]. This effect may also be due to differences in executive functioning, as a study on the CCT "hot" and "cold" versions demonstrated, but additional evidence is needed to support this hypothesis [26]. Regardless, the substantial impact of even a brief delay on impulsive responding calls into question how generalizable a participant's behavior is compared to when they are not required to wait before responding. Research using EEG or other psychophysiological measures may find that ordinarily impulsive participants may not behave very differently during an experiment compared to non-impulsive participants due to this delay [25]. Consequently, the "hot," non-delayed APRT condition should be used in future studies that correlate self-report measures with APRT performance due to this attenuation in the "cool" delay condition.

The implications of this delay effect are not completely negative. For example, some have proposed that psychopathic individuals have a cognitive processing deficit compared to non-psychopathic individuals, wherein such individuals fail to integrate immediate feedback information when focused on a risk taking task (such as a card game where they try to win money and avoid losses) and thus fail to adapt their behavioral strategy effectively [64]. However, this deficit has been shown to disappear when a delay (similar to the one imposed in the current study) is enforced before allowing the next behavioral choice [33]. Thus, the results of the current study indicate that this effect may be generalizable to non-psychopathic populations such that that most individuals, regardless of their personality traits, behave less impulsively when forced to wait before making a risky decision.

## Limitations and future directions

There are a few limitations to consider when interpreting the results of the current study. The first is the limited engagement of many of the participants in the study, who were all undergraduates from an ethnically diverse and relatively unselected student body. Such samples are typically above-average in risk taking compared to traditional community or more selective undergraduate samples [65]. We initially considered excluding participants who had zero Go Presses for half or more of the APRT trials, but we recognized that this would have eliminated around a third of the sample. Thus, we retained these participants to prevent eliminating a subgroup of participants that may have been genuinely cautious, which also kept our study powered to detect the GEE main effect sizes we anticipated from pilot data without that exclusion. Furthermore, the inclusion of these participants makes our sample more representative of the population distribution of risk taking propensity. Physical risk taking carries the risk of death in real-world scenarios, whereas other forms of risk taking do not. In that light, low response rates are not necessarily surprising given APRT's subject matter compared to that of BART or the card-based tasks. Nevertheless, the high number of zero-scored trials likely skewed the

means for all four APRT outcome scores, and further studies with larger, more diverse samples are needed to explore this non-responsive subgroup further. Changes to the APRT parameters (e.g., shrinking the gap between high/low reward magnitude to minimize satisificing) may also result in increased responding. Additionally, parametrically varying the length of the delay would allow a precise mapping of the attenuations of APRT behavior's correlations with self-report as response sets shift from representative "hot" to non-representative "cool".

The other limitations concern the APRT trials themselves. First, the probabilities of success and failure were communicated to participants through a border coloring scheme. Consequently, the task presumes participants recall each border/color combination when making selections during each trial. Failure to recall these correctly could lead to noisier behavior, as differences in scores may be influenced by differences in memory rather than risk taking. Though a potential solution would be to simply display the probabilities numerically to participants during each trial, we elected not to do this to avoid crowding the screen with information and overwhelming participants' selective attention. A future study could test this option and determine if APRT performance differs based on how these probabilities are presented (border/color vs. explicitly numeric).

A second issue with the APRT trials is that each of the 64 trials represents a unique combination of the within-subject manipulations. This means that more complex interaction effects may exist, which were not examined due to the difficulty (if not impossibility) of conceptually explaining them in ecologically useful terms and the lack of necessary power to discern these effects [66]. Adding additional trials so that each combination of factor levels is repeated several times (e.g., 128 or even 256 trials) would allow for greater assessment of the reliability of observed effects. However, acquiring the required number of stimuli (i.e., more open-source pictures of the very specific scenarios), as well as participant fatigue (as APRT would likely often be administered as part of a larger study) limit the viability of this solution. Alternatively, simulated data sets may allow for reliability testing without having to increase the number of trials. Specifically, a model recovery analysis would assist in revealing the necessary effect size required to reliably observe a set of given effects across multiple groups of participants. Future studies should therefore conduct such analyses to confirm the reliability of the current observed effects in addition to replicating these effects in additional samples.

Our decision to only examine two-way interactions (despite the possibility to examine up to five-way) is reflective of the greater issue of tension between simplicity and complexity in experimental design. Simpler, stratified versions of APRT (i.e., a version where only injury probability or magnitude is manipulated) would provide more efficient and interpretable models of the main effects of risky decision-making factors. However, they would be incapable of modeling potentially meaningful (and perhaps more impactful) interaction effects that influence behavior. Examining two-way interactions within a task that could potentially contain five-way interactions can be seen as a compromise between these conflicting needs for parsimony and precision. Thus, future studies should work to determine if the presently observed two-way interactions within APRT are robust and of practically meaningful effect size, as the degree to which these interactions represent large enough real-world effects requires further investigation to justify the complex design of APRT [66].

The current version of APRT also omitted forms of risk taking that may have been associated with disinhibition instead of thrill and adventure seeking (like drug use or drag racing), which limits the generalizability of the risks this task assesses. Future work could replace pictures of physical risks with depictions of other risky activities to examine whether populations who participate in specific kinds of deleterious behaviors show altered behavior in this task. Additionally, the static pictures and corresponding button-press responses are convenient, easily implemented abstractions of real-world risk taking scenarios and behavior. More

complex simulations, such as virtual reality with coding schemes for actual behavioral responses, would provide a more realistic task than the present version. However, this procedure would require greater technical proficiency to design and would also likely result in reduced implementation due to hardware costs. Furthermore, the absence of an external reward for achieving a high APRT score may have decreased participant engagement among individuals with low to moderate physical risk taking propensity. In this light, there is little external benefit (e.g., monetary or other prize) that justifies adopting a risky behavioral style within the task, though this is true of all laboratory studies that do not incentivize participant performance.

In sum, APRT provides a comprehensive assessment of the elements of risk across a variety of different physically risky activities within a novel conceptualization of risk taking itself. In future work, APRT may shed light on whether certain populations (such as first responders and extreme sports participants) are particularly adept at making decisions under conditions of physical risk. Having a behavioral measure of physical risk taking will also advance studies of when risk taking goes awry, such as when individuals come to the emergency department for their physical injuries [67]. However, additional studies using APRT are implementing it in applied settings. Specifically, the real-world criterion validity of APRT was not examined by the current study, so whether APRT scores correlate with meaningful criteria (e.g., substance use scores in a population of rehab facility residents, decision-making in military special forces) awaits confirmation. Future studies should also directly compare APRT to other laboratory risk taking tasks (e.g., BART, CCT, IGT) to establish the incremental validity and practical advantages of APRT. Such investigations would bolster the real-world importance of the APRT and build on our initial promising investigation of the APRT's convergent and divergent validity.

## Supporting information

**S1 Table. Summary of APRT parameter implementation across 64 trials.** Numbers in interior cells indicate the number of trials containing those specific APRT conditions.
(DOCX)

**S2 Table. Summary of descriptive statistics for APRT difference scores.**
(DOCX)

**S3 Table. Summary of descriptive statistics for self-report measures.**
(DOCX)

**S4 Table. Pearson correlations between APRT Remaining Health scores and self-report measures.** $^{*}p_{adj} < .05.$ $^{**}p_{adj} < .01.$ $^{***}p_{adj} < .001.$
(DOCX)

**S5 Table. Pearson correlations between APRT Injuries scores and self-report measures.** $^{*}p_{adj} < .05.$ $^{**}p_{adj} < .01.$ $^{***}p_{adj} < .001.$
(DOCX)

**S1 Fig. Two-way interaction between Reward Magnitude and Injury Probability on APRT Points score.**
(DOCX)

**S2 Fig. Two-way interaction between Injury Magnitude and Picture Category (Disaster-Hero) on APRT Go Presses score.**
(DOCX)

**S3 Fig. Two-way interaction between Reward Magnitude and Reward Probability on APRT Go Presses score.**
(DOCX)

## Author Contributions

**Conceptualization:** Edward A. Smith, Stephen D. Benning.

**Data curation:** Edward A. Smith, Stephen D. Benning.

**Formal analysis:** Edward A. Smith.

**Methodology:** Edward A. Smith, Stephen D. Benning.

**Project administration:** Edward A. Smith.

**Supervision:** Stephen D. Benning.

**Writing – original draft:** Edward A. Smith.

**Writing – review & editing:** Edward A. Smith, Stephen D. Benning.

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
