## [Decision Letter · Decision Letter 0]

23 Mar 2021

PONE-D-20-37198

The Assessment of Physical Risk Taking: Construct Validation of a New Behavioral Measure

PLOS ONE

Dear Dr. Smith,

Thank you for submitting your manuscript to PLOS ONE. After careful consideration, we feel that it has merit but does not fully meet PLOS ONE’s publication criteria as it currently stands. Therefore, we invite you to submit a revised version of the manuscript that addresses the points raised during the review process.

I have received two reviews of the manuscript entitled “The Assessment of Physical Risk Taking: Construct Validation of a New Behavioral Measure” that you recently submitted to PLOS One. I am very glad to have received comments and evaluations of your manuscript from two researchers who are knowledgeable and experts in the area of decision-making and risk-taking. As you will see, the reviewers have offered many detailed points and constructive suggestions on how to improve the manuscript.

I have read the manuscript prior to receiving these reviews in order to have an independent perspective and then again with the reviews in hand. I found myself agreeing with the reviewers in almost every point and recommendation they raised. All of us agree that the work tackles an interesting topic. The paper is also well-written and structured. It is also very important that you've made available all data and relevant files .

However, at the same time the reviewers have raised concerns that prevents me from accepting the paper in its current form. I share all of these concerns; However, I believe that you can address these concerns in a revision, so I would like to encourage you to submit a revision. The main points relate to the main objective of the current work (which is rather unclear or unspecified) and how it contributes (and extends) existing research and literature in the area of risk assessment and preferences. In addition, there are some open questions about certain methodological features of the APRT and the data analyses that support the main conclusions.

What I expect is that you will take each of the concerns seriously and address these concerns in two ways. First, where possible, you should make changes in the manuscript to correct shortcomings that the reviewers have identified. (If there are comments that you do not find to be correct or apt, you should still consider that the incorrect perception is something that you might expect in other readers, so it would be helpful to take steps for the paper to anticipate such misperceptions and add clarifications in the text to prevent them.) The goal is to make your paper as accurate, scientifically responsible, interesting, and accessible to a wide range readers and researchers. 

The second way that I would like you to respond to the reviewers is to put a lot of effort into a careful cover letter that goes through the comments point by point, explaining how you addressed each comment and, if you disagree with a comment, why you disagree (and, if possible, how you altered the writing in anticipation that other readers might have similar concerns).

We look forward to receiving your revised manuscript.

Kind regards,

Emmanouil Konstantinidis, Ph.D.

Academic Editor

PLOS ONE

2. Please include a caption for figure 1.

Reviewers' comments:

Reviewer's Responses to Questions

**Comments to the Author**

1. Is the manuscript technically sound, and do the data support the conclusions?

Reviewer #1: Partly

Reviewer #2: Yes

2. Has the statistical analysis been performed appropriately and rigorously? 

Reviewer #1: Yes

Reviewer #2: Yes

3. Have the authors made all data underlying the findings in their manuscript fully available?

Reviewer #1: Yes

Reviewer #2: Yes

4. Is the manuscript presented in an intelligible fashion and written in standard English?

Reviewer #1: Yes

Reviewer #2: Yes

5. Review Comments to the Author

Reviewer #1: This paper present a novel behavioral task aimed at assessing "physical risk taking". The authors claim that one of the task's main goals is to disentangle the relative influences of the i) probability and ii) size of the reward, as well as the iii) probability and iv) size of the loss that may result from taking a risk. I agree with the authors that improved behavioral tasks of risk taking are much needed. Moreover, the empirical study has been conducted carefully, and I appreciate that the materials and data are openly accessible. That said, I have several conceptual questions and comments, and overal the central goal and contribution of the paper remained somewhat unclear to me: Is it primarily to develop a novel measure for "physical" risk taking (if so: why exactly physical risk taking)? Is it to disentangle the potential drivers of risk taking as mentioned above (if so: why was this done in a "novel" domain such as "physical risk taking" and not using one of the more established domains)? Is it to study "multi-dimensional" risk problems (i.e., as opposed to the typical gambles, where the probability of a reward is often the inverse of the probability of a loss)? Is it to further examine self-regulation and impulsivity as drivers of risk taking (which seems to be implied by the hot / cold versions of the task)? Or is it about a combination of all of the above? Taken together, all in all I am not (yet) entirely convinced that this task lives up to the expectations, and several issues need to be addressed in order to better clarify the exposition of this research. I hope that my comments and questions below will be helpful to this end.

Introduction:

- The authors argue that "most definitions of risk focus exclusively on potential negative consequences". Although it is true that some disciplines adopt the view of risk taking as behavior with potentially harmful consequences, this general statement is not entirely correct. In economics (and economic psychology), for instance, risk is typically defined as "outcome variance" (see Schonberg et al., 2011) and hence "risk" also exist in gains-only settings.

- Relatedly, contrary to the portrayal in the introduction it is not the case that psychological theories of behavioral risk taking exclusively focus(ed) on the role of negative consequences. In fact, studies in the field of decision analysis (often using monetary gambles) have examined many risk-related phenomena purely in the gain domain (e.g., EU-theory). Consequently, the documentation of the "reflection effect" started to draw attention on differential effects of gains and losses, triggering the introduction of a reference point in prospect theory. To some extent, the relative importance of gains and losses has also been studied in the BART (which serves as an important starting point in the current paper), with respective models taking into account reward sensitivity, loss aversion, etc. (Wallsten et al., 2005).

- The authors write that "current behavioral task correlate with real-world risky behaviors broadly", but there is no reference to support this claim. In fact, accumulating empirical evidence indicates that behavioral tasks may lack external validity.

- Contrary to the authors' claim, many attempts have been made to decompose various tasks into basic behavioral components. As outlined above, this has been the case for the BART, but naturally also for the widespread monetary lotteries (i.e., the associated choice models often differentiate between the roles of gains / losses and the respective probabilities; cf. the fourfold pattern of risk). The same is true for the Iowa Gambling Task (cf. Yechiam, 2005).

- It is not quite true that the "injury probability condition is almost never considered in the BART" (p. 5; as a minor comment, it seemed a bit odd to me to use the term "injury" in the context of the BART, maybe consider using "loss" or a similar term). Specifically, the BART-models mentioned above (Wallsten et al., 2005) were specifically developed to account for (subjective) explosion probabilities (e.g., to compare to the objective explosion probabilities). Moreover, in more recent research, the impact of different [explosion] probability conditions was directly assessed with respective prompts and modeling (Schürmann et al., 2019; Steiner et al., 2020).

Behavioral task:

- The authors argue (p. 7) that previous tasks may be constrained as they consist of games without "a real-world equivalent". In line with this argument, it has been proposed in the literature that some tasks (including the BART) may be constrained in terms of their generalizability (see comment above) because they lack a representative design (Steiner et al., 2020). Thus, it would be important to introduce how / why the four settings implemented in the APRT are ecologically more relevant (e.g., encountering a tiger does not seem an overly realistic real-life experience).

- The authors claim that an advantage of the novel task is that a participant cannot simply infer the probability of a gain from the probability of a loss (or vice versa), as there are also "neutral" outcomes. Thus, can one argue that this is a three-outcome setting? If so, it would be interesting to relate that to the respective research (e.g., on monetary gambles) that has used setups other than the typical two-outcome gambles.

- Relatedly, I wondered what to make of the additional dimension of "health". It seemed to me as if the participants' had to deal with two non-fungible "currencies" in the task (i.e., maximize payoffs in terms of the points, or maximize health)? Can the authors elaborate further on this issue?

- Finally, I was also somewhat puzzled by way the information about the probabilities of success / failure was communicated to participants (the colored / striped borders). The authors argue that this procedure "protects the task from learning effects", but I could not quite follow this rationale. In fact, this creates an additional learning component exactly *because* participants have to remember (and recall) the meaning of this symbolic representation (which in itself could lead to noisier behavior). Second and under the assumption that participants could perfectly recall the meaning of the various borders, how / why would this be any different from directly depicting the probabilities?

Hypotheses / analyses:

- I found the theoretical motivation for the various hypotheses (p. 11) relatively thin. This is particularly an issue as the model implemented to test these hypotheses is very flexible (i.e., many terms and interactions; generally many exploratory analyses). Relatedly, although participants completed a number of trials, I wondered whether the various parameters are fully identified in this model. A simulation / model recovery analysis could be informative and boost the confidence in the robustness of the results.

Schonberg, T., Fox, C. R., & Poldrack, R. A. (2011). Mind the gap: Bridging economic and naturalistic risk-taking with cognitive neuroscience. Trends in Cognitive Sciences, 15(1), 11–19. https://doi.org/10.1016/j.tics.2010.10.002

Schürmann, O., Frey, R., & Pleskac, T. J. (2019). Mapping risk perceptions in dynamic risk-taking environments. Journal of Behavioral Decision Making, 32, 94–105. https://doi.org/10.1002/bdm.2098

Steiner, M. D., & Frey, R. (2020). Representative Design in Psychological Assessment: A Case Study Using the Balloon Analogue Risk Task (BART). https://doi.org/10.31234/osf.io/dg4ks

Wallsten, T. S., Pleskac, T. J., & Lejuez, C. W. (2005). Modeling behavior in a clinically diagnostic sequential risk-taking task. Psychological Review, 112(4), 862–880. https://doi.org/10.1037/0033-295X.112.4.862

Yechiam, E., Busemeyer, J. R., Stout, J. C., & Bechara, A. (2005). Using cognitive models to map relations between neuropsychological disorders and human decision-making deficits. Psychological Science, 16(12), 973–978. https://doi.org/10.1111/j.1467-9280.2005.01646.x

Reviewer #2: The manuscript “the assessment of physical risk taking: construct validation of a new behavioral measure” presents a new method of assessing physical risk taking which simultaneously varies probability and reward in more realistic contexts than existing behavioral measures. As an initial presentation, the paper is adequate in it’s rationale and uniqueness. The data presentation is fairly thorough and there is some initial evidence of construct validity.

Therefore as an initial presentation I believe the manuscript is adequate, however I believe the limitations and future directions should be more explicit on what is not in the manuscript (and given the scope of the paper probably shouldn’t be in this paper). First, construct validity is not a conclusive state but a process of accumulation that would need to continue in future studies. Second, criterion validity is absent and would be required before the measure could be used in any real-world applications. Third, a direct comparison between the APRT, BART, IGT, and other major behavioral risk-taking measures would be required to illustrate the practical advantages the APRT claims. Fourth, the issue of non-response mentioned in the limitations needs further study with larger and more diverse samples. And finally, given all of these points, it should be made clear that the APRT requires further evidence of validity and utility prior to any applied use.

6. PLOS authors have the option to publish the peer review history of their article (what does this mean?). If published, this will include your full peer review and any attached files.

Reviewer #1: No

Reviewer #2: No

---

## [Author Response · Author response to Decision Letter 0]

7 May 2021

Thank you for allowing us to revise our submission. Each and every point raised by reviewers has been addressed in our "Response to Reviewers" document. You may find all our responses within, broken down as a direct response to each point raised by reviewers.

---

## [Decision Letter · Decision Letter 1]

10 Aug 2021

PONE-D-20-37198R1

The Assessment of Physical Risk Taking: Preliminary Construct Validation of a New Behavioral Measure

PLOS ONE

Dear Dr. Smith,

Thank you for submitting your manuscript to PLOS ONE. After careful consideration, we feel that it has merit but does not fully meet PLOS ONE’s publication criteria as it currently stands. Therefore, we invite you to submit a revised version of the manuscript that addresses the points raised during the review process.

I have now received one expert review of your manuscript; the reviewer agree that you present a much improved version of the manuscript. While most of the comments have been addressed in this revision, the reviewers have suggested a few areas that the manuscript can be improved (mostly minor comments). I also believe that the manuscript will benefit from a final revision of the submitted work which addresses the reviewer's comments.

We look forward to receiving your revised manuscript.

Kind regards,

Emmanouil Konstantinidis, Ph.D.

Academic Editor

PLOS ONE

Journal Requirements:

Reviewers' comments:

Reviewer's Responses to Questions

**Comments to the Author**

1. If the authors have adequately addressed your comments raised in a previous round of review and you feel that this manuscript is now acceptable for publication, you may indicate that here to bypass the “Comments to the Author” section, enter your conflict of interest statement in the “Confidential to Editor” section, and submit your "Accept" recommendation.

Reviewer #1: All comments have been addressed

2. Is the manuscript technically sound, and do the data support the conclusions?

Reviewer #1: Yes

3. Has the statistical analysis been performed appropriately and rigorously? 

Reviewer #1: Yes

4. Have the authors made all data underlying the findings in their manuscript fully available?

Reviewer #1: Yes

5. Is the manuscript presented in an intelligible fashion and written in standard English?

Reviewer #1: Yes

6. Review Comments to the Author

Reviewer #1: The authors have carefully responded to all of my previous comments and questions, and I believe that the revised paper now does a much better job in highlighting the contributions (and limitations) of the current study. I have a number of final suggestions that may help to further strengthen the exposition of this work.

1. On p. 4 the authors argue that previous tasks do not correlate strongly with real-life behaviors and hence, that there is a need for more complex behavioral tasks. Alike, on p. 46 the authors argue that maybe even "more complex simulations" are required (i.e., to capture additional dimensions relevant to risk taking). So this is actually rather a comment than a suggestion, but personally I do not quite agree with this conclusion. In my view many tasks are already too complicated, often making it difficult to identify the exact reasons why people do (vs. do not) take risks in these tasks (especially if only few trials are collected; see also comment 3 below). An alternative and in my view more promising approach consists of actually developing simpler tasks, which may capture specific risk-related dimensions more precisely. Again, this is my personal opinion and should not prevent publication of this paper; the respective debate needs to be conducted in the field.

2. The study on the novel task aims to achieve many things at once (i.e., multiple conditions and various outcomes are compared against multiple measures and scales), and ultimately the reader may wonder how robust the reported effects indeed are. I appreciate that the authors acknowledged the exploratory nature of their research in the revision and that they sharpened their predictions where applicable. In my view, however, it could be helpful to make the exploratory nature even more explicit, in order not to raise any wrong expectations. For example, adding something like "An exploratory investigation of..." to the title and / or abstract may help setting expectations straight.

3. With comments #1 and #2 in mind, I would find it essential that future research clarifies the identifiability and reliability of the parameters estimated in this task. Again, many things are being manipulated and measured here – but are the various dimensions indeed clearly separable, and is the number of trials sufficient to identify reliable response patterns? As outlined in my previous review, a model recovery analysis could be very instrumental to this end. Briefly, the idea is to implement various data-generating processes (e.g. simulate groups of people who take risks for different reasons in different conditions), to then estimate the models and evaluate whether the implemented effects can be recovered (or how strong these effects would need to be in order to be recovered reliably). The authors responded that they are not confident implementing such an analysis - fair enough, but it would be very helpful to have at least an explicit discussion of the option to run a parameter recovery study in the future, to thus clarify the reliability of the extracted parameters / effects.

4. Finally and also with expectation management in mind, I wondered whether it would not be more accurate / informative to use "simulation of" instead of "assessment of" physical risk taking in the name of the task. The latter implies that some concrete real-life behaviors are assessed, which is naturally not quite the case as the authors also acknowledge in the paper.

7. PLOS authors have the option to publish the peer review history of their article (what does this mean?). If published, this will include your full peer review and any attached files.

Reviewer #1: No

---

## [Author Response · Author response to Decision Letter 1]

18 Sep 2021

Please see attached "Response to Reviewers" at end of submission package. All comments are addressed in detail there.

---

## [Editor Report · Decision Letter 2]

7 Oct 2021

The Assessment of Physical Risk Taking: Preliminary Construct Validation of a New Behavioral Measure

PONE-D-20-37198R2

Dear Dr. Smith,

We’re pleased to inform you that your manuscript has been judged scientifically suitable for publication and will be formally accepted for publication once it meets all outstanding technical requirements.

Kind regards,

Emmanouil Konstantinidis, Ph.D.

Academic Editor

PLOS ONE

---

## [Editor Report · Acceptance letter]

11 Oct 2021

PONE-D-20-37198R2 

The Assessment of Physical Risk Taking: Preliminary construct validation of a new behavioral measure 

Dear Dr. Benning:

I'm pleased to inform you that your manuscript has been deemed suitable for publication in PLOS ONE. Congratulations! Your manuscript is now with our production department. 

Kind regards, 

on behalf of

Dr. Emmanouil Konstantinidis 

Academic Editor

PLOS ONE